# Eco-Friendly, Biodegradable Starch-Based Packaging Materials with Antioxidant Features

**DOI:** 10.3390/polym16070958

**Published:** 2024-04-01

**Authors:** Dagmara Bajer

**Affiliations:** Faculty of Chemistry, Nicolaus Copernicus University in Toruń, Gagarina 7, 87-100 Toruń, Poland; dagmara@umk.pl

**Keywords:** starch, biodegradable packaging materials, antioxidant properties

## Abstract

Due to the extensive application of petroleum-based plastics as packaging materials and problems related to their degradation/recycling, developing new solutions in the field of novel biopolymer-based materials has become imperative. Natural substitutes for synthetic polymers (starch, cellulose, chitosan) require modifications that enable their processing and provide them with additional properties (i.e., mechanical strength, controlled biodeterioration, antimicrobial and antioxidative activity). The antioxidant activity of natural packaging materials still requires further investigation. In this research paper, novel materials used for packaging perishable food susceptible to oxidizing agents were designed from potato starch (NS) reinforced with antioxidants such as dialdehyde starch (DS) and caffeic acid (CA)/quinic acid (QA). The use of spectroscopic techniques (ATR-FTIR, Raman) and X-ray diffraction allowed the examination of the chemical structure and arrangement of the blend and confirmed the component interactions. The film surface was examined by AFM. DS, functioning as a cross-linker, enhanced the film barrier as well as the mechanical and thermal properties, and it promoted starch amorphization when blended with other antioxidants. The antioxidant activity of caffeic acid was greater than that of quinic acid. Dialdehyde starch improves elasticity, whereas acids (particularly caffeic acid) influence film stiffness. A high susceptibility to biodegradation is valuable for potential eco-friendly packaging applications.

## 1. Introduction

Petroleum-based materials, which are prevalent in the packaging industry, pose a threat to the natural environment due to their nonbiodegradability and complicated recycling process. Designing biodegradable substitutes for synthetic plastics based on polysaccharides (cellulose, starch, and chitosan) has become a necessity nowadays and thus has become the primary goal of packaging industry research [1,2,3]. The most essential requirements for such packaging materials necessary to maintain food quality are mechanical strength, planned durability, and biodegradability.

Despite its many advantages, including wide availability and nontoxicity, native starch requires physical or chemical modification and plasticization to become a competitive raw material for packaging purposes. The general differences in the chemical structure and arrangement of starch (amylose to amylopectin ratio) resulting from its botanical origin cause specific processing problems. For example, a higher amylose content improves the mechanical and barrier characteristics of a starch-based material [4,5]. Other large-scale industrial applications may require higher mechanical resistance and stability (e.g., in the production of disposable tableware and agricultural films). Therefore, such products should be characterized by their stability during use, water resistance, gas permeability, good mechanical strength, and biodegradability. To meet the above requirements, the functional properties of starch need to be improved. This can be achieved by modifying starch with dialdehyde starch. Dialdehyde starch easily cross-links with starch, improving the homogeneity and polymeric material strength. Higher moisture content (MC = 7–12%), water solubility (WS = 4–8%), and tensile strength (TS = 2–3 MPa) were confirmed for the starch/dialdehyde starch film than for the MC (~11–14%), WS (~8–19%), and TS (~0.5–0.7 MPa) films, which were determined for starch/silica blends [6].

Due to the presence of reactive carbonyl groups, dialdehyde starch appears to be a multifunctional antimicrobial and antiviral agent. Additionally, it limits moisture absorption and improves the antioxidant properties of the final material [7,8]. The functional properties of starch and dialdehyde starch mixtures can be further improved by the addition of biologically active acids of natural origin that are safe for humans and the environment. It can be caffeic or quinic acid, characterized by antibacterial and antioxidant properties. Quinic acid (QA, 1,3,4,5-tetrahydroxycyclohexane-1-carboxylic acid) is naturally available in fruits, vegetables (apples, peaches, pears, plums, carrots), and other plants (cinchona, coffee, tobacco). Caffeic acid (CA, 3,4-dihydroxycinnamic acid) is one of the most abundant hydroxycinnamic acids present in coffee, cereals, fruits, vegetables, legumes, and nuts [9,10,11]. The numerous healing properties of both acids are valuable for the medical, pharmaceutical, and cosmetic industries. Moreover, their antioxidative and antimicrobial activities can also be valuable for the food and food packaging industry [12,13]. These studies are a continuation of previous studies on obtaining starch materials with antioxidative properties, where ascorbic acid and caffeine were used as modifiers [7].

This research aimed to obtain and characterize novel starch/dialdehyde starch blends with natural and safe caffeic/quinic acid. Such a modification was supposed to improve the properties of starch membranes, particularly by strengthening their structure and improving their mechanical and antioxidant properties. The subject of interest was to explain the impact of particular components on the final properties of the composite (e.g., to confirm expectations of maintaining the antioxidant effectiveness of acids when bonded with starch). Moreover, the surface and physicochemical characteristics of the films, such as their thermal behavior, water permeability, and hydrophilicity, were investigated. The Oxi-Top test verified their biodegradation, confirming the environmental and chemical safety of the obtained blends. The presented materials and the characteristics of their properties are part of a broader topic concerning the modification of highly hydrophilic starch for industrial applications. The proposed starch derivative enriched with antioxidants (caffeic and quinic acids) can be a valuable packaging material for protecting products (e.g., food, cosmetics, medicines) from premature aging due to oxidation [14,15,16].

## 2. Materials and Methods

### 2.1. Materials

Native potato starch (NS) with an amylose content of 26% was purchased from Cargill Poland. POCh S.A. (Gliwice, Poland) supplied technical glycerin (96% purity), sodium periodate (regent grade), NaOH micropills (reagent grade), HCl (35–38%, reagent grade), and ethanol (96%, reagent grade). Sigma–Aldrich (Poznań, Poland) supplied anhydrous caffeic acid powder (CA), quinic acid (QA), 2,20-diphenyl-1-picrylhydrazyl radical-DPPH (95%), and 6-hydroxy-2,5,7,8-tetramethylchroman-2-carboxylic acid-TroloxTM (TE) (97%).

### 2.2. Preparation of Dialdehyde Starch

Dialdehyde starch (DS) was prepared by acid hydrolysis and oxidation following the procedure described below [7]. A total of 7.5 g of potato starch was added to a solution of NaIO_4_ (4.5 g) in HCl (C_HCl_ = 0.6 mol/L, V_HCl_ = 75 mL) and mixed with a magnetic stirrer at 35–40 °C in a water bath for 2 h. The slurry was then filtered, washed first with distilled water and then with acetone, and left to dry at 21 ± 1 °C. The aldehyde group (C=O) concentration (40%) was determined by the titration method according to a previously described procedure [16].

### 2.3. Film Preparation

Potato starch film (NS), which was used as the reference sample, was prepared according to the following procedure: 6 g of starch, 2 g of glycerin plasticizer, and 150 mL of distilled water were mixed at 600 rpm (with a magnetic stirrer) for 30–40 min at 65 ± 5 °C. Subsequently, the starch suspension was left on PVC pans to dry at room temperature (20–22 °C).

To prepare samples of potato starch with caffeic acid or quinic acid (NS/CA, NS/QA), the reference starch solution (procedure described above) was cooled to 25 °C and then mixed with a solution of caffeic acid (CA) or quinic acid (QA) (0.3 g of acid/10 mL of distilled water) for 5 min with a magnetic stirrer (600 rpm). These prepared suspensions were poured into PVC and left to undergo solvent evaporation and film formation.

The last sample set was enriched with dialdehyde starch (DS) and acids. First, 0.8 g of DS was placed in 0.1 M NaOH aqueous solution (pH adjusted to 10) and mixed at 600 rpm under heating in a water bath (70 °C) until DS dissolved. Subsequently, the DS and acid (caffeic/quinic) solutions were transferred to the basic starch solution and mixed for 5 min. The following compositions were prepared: NS/DS/CA—the film composed of native potato starch, dialdehyde starch, and caffeic acid—and NS/DS/QA—the film of native potato starch with dialdehyde starch and quinic acid.

As a reference sample, an NS/DS film was also prepared.

### 2.4. Raman Spectroscopy

Raman spectroscopy was performed with a Senterra Raman confocal dispersion spectrometer (Bruker Optik, Ettlingen, Germany, Woodlands, TX, USA). Spectra (in the spectral range of 90–3500 cm^−1^) were collected at room temperature with a power range of 1–100 mW. OPUS 7.5 software (Bruker Optics Instruments, Ettlingen, Germany) was used for analysis of the results.

### 2.5. FTIR-ATR Spectroscopy

FTIR-ATR spectra were recorded on a Spectrum TwoTM instrument (PerkinElmer, Waltham, MA, USA) with an ATR device with a diamond crystal. All the spectra were recorded with the following conditions: a resolution of 4 cm^−1^, a range of 400–4000 cm^−1^, and a scan number of 60. The spectra were analyzed with Omnic 9.2.41 Thermo Fisher Scientific Inc. software (Waltham, MA, USA). The hydroxyl index (R_OH_), which describes the hydrophilicity of the film, was determined according to Formula (1):R_OH_ = I_OH_/I_CH3_,(1)
where:I_OH_—the integral intensity of the hydroxyl absorption band in the spectral range of 2900–3600 cm^−1^, andI_CH3_—the integral intensity of the methylene absorption band in the range of 2800–2900 cm^−1^.

### 2.6. Powder Diffraction (P-XRD) Measurements

X-ray powder diffraction analyses were conducted with an X’Pert PRO diffractometer (Malvern Panalytical, Almelo, Holland) using nickel-filtered CuKα radiation (1.5240 Å wavelength). Spectra were collected at 2θ values ranging from 5 to 60°. The crystallinity degree (X_c_) was estimated as the ratio between the total area of the crystalline peaks (X_cr_) and the total diffractogram area (sum of the amorphous and crystalline halos, X_am_ + X_cr_) according to Formula (2):X_c_ = X_cr_/(X_am_ + X_cr_); % (2)

### 2.7. Thermal Properties (Thermogravimetry)

Thermogravimetric analysis, conducted with a TA Instrument SDT2960 simultaneous analyzer (New Castle, DE, USA), allowed simultaneous recording of TG, DTG, and DTA curves. Five to ten milligrams of sample was heated from 20 °C to 600 °C at a rate of 6 °C/min in a nitrogen atmosphere.

### 2.8. Mechanical Tests

The mechanical strength was verified with a mechanical testing machine with tensile grips (EZ-Test SX Texture Analyser, Shimadzu, Kyoto, Japan). The results were analyzed with Trapezium X software (version 1.4.5, Shimadzu, Kyoto, Japan).

The analyzed samples were cut into paddle shapes (50 mm long and 4.5 mm wide, the average film thickness was 0.176 ± 0.023 mm) and stretched to break between the machine clamps at a velocity of 10 mm/min. The Young’s modulus (E), stress at break (σ_b_), and elongation at break (ε_b_) values were established based on at least five measurements for each type of sample.

### 2.9. Water Vapor Transmission Rate (WVTR)

The water vapor transmission rate through the starch-based films was examined according to the standards [17]. Five grams of silica gel dried to a constant weight was placed in the measuring containers and closed with a lid with a hole of 3 cm × 3 cm, which formed the testing area of 9 cm^2^. The film samples (with a thickness of approximately 0.25 ± 0.01 mm) were placed in lid openings to allow water vapor penetration and absorption by the silica gel inside the cup. Measurements were carried out for 7 days in a Q-CELL 60 incubation chamber (POL-LAB sp. z o.o., Wilkowice, Poland) under constant conditions: a temperature of 23 °C and a relative humidity of 70%. The measurements were determined in triplicate for each type of sample.

The WVTR (g/(m^2^·h)) was established as follows (Formula (3)):WVTR = Δm/A·t;(3)
where Δm is the weight gain of the cup (g), A is the testing sample area (m^2^), and t is the time of the experiment (h).

### 2.10. Antioxidant Capacity (DPPH Radical Scavenging Assay)

The antioxidant activity was established by a DPPH radical scavenging assay according to a method described in the literature [18].

First, 0.5 mL of fresh DPPH ethanolic solution at a concentration of 300 µmol/mL, 2.0 mL of ethanol, and 2 µL of the analyzed sample were placed in a measuring cuvette, stirred for 0.5 min, and incubated in the dark for 15 min. In the next step, the absorbance was measured at 517 nm using a UV-1601 PC spectrophotometer (Shimadzu Corporation, Japan). The results were read from the linear relationship of the Trolox standard solution concentration (µmol/mL) vs. DPPH (%). Formula (4) was used to calculate the radical scavenging activity of DPPH:DPPH = [(A_control_ − A_sample_)/A_control_]·100%(4)
where:
A_control_—absorbance of DPPH radical,A_sample_—absorbance of DPPH radical and the analyzed solution.

The presented results are the averages from 3 measurements.

### 2.11. Atomic Force Microscopy (AFM)

Atomic force microscopy images were obtained with a MultiMode NanoScope III (Veeco Metrology Inc., Goleta, CA, USA) instrument equipped with a silicon probe (Veeco) in tapping mode in air. The roughness parameters, arithmetic mean (R_a_), and root mean square (R_q_) were determined for the film after scanning a sample area of 5 μm × 5 μm at room temperature.

### 2.12. Biodegradation in Soil

The degree of biodegradation was determined by respirometric tests using the WTW OxiTop^®^—Control 110 (Xylem Water Solutions Polska Sp. z o.o., Warsaw, Poland) apparatus set to the “Pressure p” mode. Starch-based membranes suspended in the selected environment were incubated in a hermetically sealed 1 L glass jar for 28 days at 20 °C. The mass of the soil was 100 g, and the mass of the film was 1 g (the foils were previously cut into pieces with an area of approximately 0.5 cm^2^). The pressure inside the vessel (the basis for determining the biological demand for oxygen) was measured once every 24 h. The reference was endogenous respiration (only fresh soil was used). All the samples were analyzed in triplicate. The biological oxygen demand (BOD) during sample biodegradation was calculated as mg O_2_/kg fresh weight of soil.

## 3. Results and Discussion

Spectroscopic techniques confirmed interactions and chemical bond formation between starch and the blended additives. The Raman characteristic starch bands at 851 cm^−1^, 1052 cm^−1^, 1071 cm^−1^, and 1462 cm^−1^ were related to C-C ring breathing and stretching vibrations, C-O-C stretching, CH_3_ rocking, and the stretching vibrations of the CH, CH_2_, and C-O-H bonds, respectively [19,20] (Figure 1). The maxima between 900 and 1500 cm^−1^ corresponding to CH-OH stretching are also typical for both organic acids [21]. The absorption bands observed at 1500–1700 cm^−1^ are known to be associated with water O–H stretching vibrations in the NS/QA and NS/DS/QA spectra.

Raman spectra exhibit a strong decrease in intensity in the C-H stretching mode with a maximum at ~2900 cm^−1^ and in a spectral range characteristic of O-H groups (3100–3700 cm^−1^) in starch/acid spectra compared to the spectrum of native starch [9]. These bands further decreased for the blends with dialdehyde starch, confirming the cross-linking action. The starch-attributed maximum at 1669 cm^−1^ correlates with the amylopectin concentration [22]. This peak almost disappeared when starch was blended with quinic/caffeic acid and DS, which may be attributed to the increase in the degree of starch substitution.

The caffeic acid in NS/CA is characterized by a low intense peak at 1242 cm^–1^, corresponding to the vibration of the hydroxyl in the benzene ring, which further disappeared in the NS/DS/CA spectrum. The other strong maxima correlate with the carbonyl stretching of the benzene moiety (in caffeic acid) and the C-C stretching of benzene and of acyclic chains at 1642 cm^−1^ and 1619 cm^−1^, respectively. The intensity of the carbonyl band decreased due to the formation of new bonds in the reaction of the carbonyl groups of dialdehyde starch/caffeic acid and hydroxyl groups of native starch (caffeic ester formation) (the maximum at 3375 cm^−1^ disappeared, Figure 1). In addition, the vibration maxima at 1185 and 1535 cm^−1^ of the ring-conjugated C=C stretching in the NS/CA of coniferaldehyde and sinapaldehyde disappeared in the NS/DS/CA spectrum [23]. Raman spectroscopic analysis also confirmed the mutual interaction between the NS/QA and NS/DS/QA mixtures. The spectral regions at 1500–1200 and 1200–300 cm^−1^ are assigned to –CH_2_ (wagging and rocking modes), carboxyl groups, and –OH deformation, respectively, corresponding to the cyclohexane ring of quinic acid [9]. Blending with dialdehyde starch caused all the signals characteristic of NS/QA to decrease in intensity (maxima at 1415, 940, 852, and 483 cm^−1^). Additionally, some maxima disappeared (at 1457, 1149, 1106, 766, and 671 cm^−1^).

The peak characteristic of the C-O stretching modes of quinic acid appears at 1084 and 1052 cm^−1^. Furthermore, a medium-intensity signal at ~850 cm^−1^ and a low-intensity signal at ~1240 cm^−1^ (Figure 1), characteristic of the C–C and C–O bonds of the starch pyranose ring and the –CH_2_ deformation vibration of the starch backbone, are observed in starch, and they irregularly change in intensity. The band at ~483 cm^−1^ lost its intensity, which confirms the interruption of the short-range molecular order of the starch after blending with DS [24].

Furthermore, ATR-FTIR spectroscopy confirmed that there were mutual interactions between the blend components. The spectra of NS/DS, NS/DS/CA, and NS/DS/QA were characterized by high similarity to the native starch spectrum (Figure 2). The locations of the following maxima, namely, the hydroxyl stretching vibration of glucose units (the peak at 3000–3600 cm^−1^), methylene stretching (at 2933 cm^−1^), and methylene bending (1200–1500 cm^−1^), are unaltered. The new maxima that appeared at 1700 and 1742 cm^−1^ in the NS/DS spectra resulted from periodate oxidation, which led to carbonyl group creation at C2 and C3 of the dialdehyde starch anhydroglucose units. Their intensity changes and significant shifts are observed for blends composed of starch, both acids, and dialdehyde starch; however, they are the strongest for the NS/DS/CA sample (1739 cm^−1^).

The peak from 3200–3400 cm^−1^ (corresponding to the O-H groups in starch and acids) significantly decreased in intensity when the dialdehyde starch was blended with the native starch. This is due to the reaction of the hydroxyl groups of starch and dialdehyde starch, and the carbonyl groups in dialdehyde starch form new ester bonds. In the corresponding spectra for NS/CA and NS/DS/CA, a decrease in the intensity of the band at 3397 cm^–1^ (attributed to O–H attached to the benzene ring of caffeine acid as well as the hydroxyl of starch and DS) is also observed. However, the carbonyl maximum (characteristic of DS) is still intense. This suggests that the binding between NS or NS/DS and caffeic acid occurs incompletely, as observed for starch–DS–quinic acid. Additionally, the hydroxyl index (R_OH_) confirmed the slight increase in the hydrophilicity of starch modified with dialdehyde starch and both acids. The R_OH_ values increased for NS/DS/CA (4.52) and NS/DS/Q (4.48) compared to those of the NS film (3.25) and samples with acids only (3.8–3.9).

Additionally, native starch, when blended with both acids and/or dialdehyde starch, was susceptible to conformational changes. This was confirmed by the decreasing intensity of the maxima ranging from 900–1200 cm^−1^ (the maxima at ~1040 cm^−1^ and ~1020 cm^−1^ correspond to crystalline and amorphous areas in starch molecules, respectively) [7]. XRD spectroscopy also confirmed that the rearrangement in starch ordering was influenced by starch modification.

The X-ray diffraction patterns of native starch (NS) and its mixtures with dialdehyde starch/caffeic acid/quinic acid are presented in Figure 3. The signals at 2θ = ~5°, 15°, 17°, 20°, 22°, and 24° correspond to potato starch B-type crystallites (Figure 3) [9]. The XRD analysis allowed the calculation of the crystallinity index (X_c_), which is shown in Table 1. The crystallinity of the NS sample (starch plasticized with glycerol) was 26%, which confirms its semicrystalline nature. All the additives affected starch amorphization and reorganization of its crystallographic structure (Figure 3a). Such transformation and loss of arrangement (~13%) were most common for starch blended with DS. The formation of new hydrogen bonds between interweaved amylose chains and glycerol plasticizers may explain this result [25]. This was also reflected in the decrease in the intensity of the characteristic signals of starch (when compared to those of NS).

The strong DS peak at 2θ ≈ 19° confirms the oxidation of starch and cleavage of the C2–C3 bond in the anhydroglucose ring. This is accompanied by the greatest decrease in crystallinity and fluctuations in the ordering in the long range (Figure 3b) [26]. Mechanical strength tests confirm these results. The degree of crystallinity (X_c_) varied from ~13.5 to 15.5% for starch blended with both dialdehyde starch and quinic acid or caffeic acid. Both acids limit the influence of dialdehyde starch on structural ordering. Moreover, amylose–amylopectin cocrystallization and amylose crystallization into single helices may explain the observed X_c_ variation. The XRD results (noticed amorphization) confirmed the increase in the barrier properties of the modified starch film and the observed decrease in the WVTR (Table 1). This behavior was slightly stronger for starch modified with quinic acid than for starch modified with caffeic acid. Moreover, the barrier properties were further enhanced by adding dialdehyde starch, which proves the good homogeneity and integrity of these films. This confirms the formation of a more compact structure with starch. According to the ATR-FTIR results, samples with dialdehyde starch are characterized by greater moisture absorption, which is not synonymous with its permeability. DS (similar to the influence of acids) slightly reduces the permeability of starch-based films to moisture. This led to the conclusion that cross-linking and hemiacetal formation occurred, which influenced the reduction in the WVTR. The increase in barrier properties may result from the low permeability and good dispersion of the filler in the starch film, which was also observed in previous studies [7]. The water barrier characteristics are comparable to the WVTRs of commonly used packaging polymers, with thicknesses ranging from 0.05~0.1 mm and WVTR values ranging from 0.04–25 g/m^2^ h. Such values are needed in food packaging applications to avoid dehydration of fresh food [27].

It may be concluded that the homogeneity of the film directly influences the barrier properties and determines the strength of the polymer material. The mechanical strength, characterized by Young’s modulus (E, MPa), stress at break (υ_b_, MPa), and elongation at break (ε_b_, %) are summarized in Table 1. Compared with those of the native starch film, the Young’s modulus, which defines the film stiffness, increased significantly for the samples containing acids and both acid and dialdehyde starch. This behavior is opposed to that of the NS/DS sample, where a decrease in material stiffness was observed. The elongation at break increases for NS/DS and confirms its highest flexibility. The effect of dialdehyde starch on NS/acid mixtures is markedly different, which may result from differences in homogeneity and the opportunities for new bond formation. The films of starch modified with CA were characterized by the most significant decrease in mechanical parameters. The caffeic acid molecule may constitute a particular steric hindrance and limit the possibility of intermolecular hydrogen bond formation with starch, resulting in a decrease in the strength and stiffness of the films. This may be a consequence of phase separation between the blend components. NS/DS/CA formed the most brittle film, characterized by the highest sensitivity to breakage (when compared with NS/CA and blends with quinic acid, which forms a more rigid material with higher fracture toughness). This indicates some difficulties in intermolecular starch–caffeic acid–DS binding, whereas intramolecular hydrogen bonding in starch molecules may be preferable. However, opposite effects and an increase in flexibility were already observed for the NS/DS/QA sample. Compared with the NS/DS/CA and NS/CA mechanical parameters, the films with quinic acid had better mechanical properties, resulting in greater resistance to rupture. Despite some differences in strength parameters, they are roughly on the same order of magnitude as the most popular synthetic materials (PE or PS) and biopolymers and biocomposites designed for food packaging. Some examples of such products, with their short characteristics, are listed in Table 2.

Furthermore, the DPPH radical scavenging activity was used to evaluate the antioxidant activity, which was significantly different for samples treated with different modifiers (Table 1). The starch linked with dialdehyde starch (NS/DS) exhibited low DPPH scavenging activity (3%). A slightly higher result was obtained for starch modified with quinic acid (approx. 5%), and further enhancement (up to ~7%) was noticed for the composite with dialdehyde starch (NS/DS/QA). These results indicate the unexpectedly low antioxidant potential of these films. On the other hand, a marked increase in DPPH scavenging activity (up to 80%) was observed for starch mixed with caffeic acid, while a decrease (69%) was detected for the sample modified with dialdehyde starch (NS/DS/CA). This is attributed to aldehyde groups in DS acting as donors of hydrogen atoms and participating in radical scavenging. Furthermore, some DS-carbonyl groups were probably consumed for conjugation with caffeic acid, which lowered the antioxidant activity [34]. The results obtained for samples with caffeic acid are auspicious and higher than the results obtained for starch films with vitamin C and with/without dialdehyde starch, which were 44.6% (starch/ascorbic acid film) and 65% (starch/dialdehyde starch/ascorbic acid), respectively [7]. These findings are also comparable to research conducted on gelatin films supplemented with Ginkgo biloba extract (GBE), which is rich in antioxidant flavonoid compounds. When GBE was introduced into gelatin films at concentrations of 0–5 g/100 g, the radical scavenging increased from 24.7% to 85.6%, making this film attractive for food packaging [35]. Increased antioxidant activity was also observed for edible chitosan/algae phenolic extract (CS/CAEE), where the moderate antioxidant activity of 21.01% for the reference CS film increased to approximately 48% with increased concentrations of CAEE to 28% [36]. The gallic acid (GA)-grafted chitosan showed a DPPH scavenging ability of 89.5% compared to the result obtained for nongrafted, native chitosan, which had an insignificant value of this parameter equal to 9.4% [37].

As noted using spectroscopic techniques, cross-linking and hemiacetal formation are highly probable in the DS mixture with the other components, which was also confirmed by the thermal behavior of the samples (Figure 4). The starch-based film exhibited a two-stage degradation process. The weight loss in the first heating stage (temperature range of 100–250 °C) reflects the adsorbed and bound moisture in the matrix. The onset of the main stage of thermal decomposition, attributed to a 53–65% mass loss, occurs at a temperature of approximately 240 °C for all the samples. The most pronounced loss in thermal stability was observed for mixtures including acids (caffeic or quinic) and DS. The NS/QA and NS/CA samples showed thermal stabilities similar to that of the NS sample (with the maximum thermal degradation occurring at 306 °C).

Notably, the thermal stability of the dialdehyde starch decreases as the aldehyde concentration increases. This resulted in a decrease in temperature at the maximum rate of decomposition of the starch sample (T_max_ determined for the NS/DS blend was 303 °C) compared with the T_max_ of the starch film (~308 °C) [7,14,38]. The value of T_max_ shifted from 302–306 °C (NS/acid films) to 284–286 °C (for NS/DS/acids). The char yield at 638 °C supports this conclusion (Figure 4). These values were higher for samples containing dialdehyde starch, confirming intra- and intermolecular bonding between the mixture components, resulting in more stable agglomerate formation.

Differences in the surface morphology of the starch films are also related to the composition of the mixture. They can be estimated by the roughness parameters of the mean roughness (Ra) and mean square roughness (Rq). The AFM images showing the surface topography of the starch-based films are presented in Figure 5. Compared with those of the other samples, the surface of the native starch film displayed finer corrugation and lower Ra and Rq values (Table 1). Although the blend topographies do not differ significantly, the NS/DS/QA blend has the most uneven surface, while the NS/DS/CA sample is a material with a relatively smooth and compact surface and roughness parameters below 10 nm, which confirm the good physical intercalation of additives in this starch blend.

The biodegradation of packaging materials can be monitored by measuring the biological oxygen demand (BOD) of microorganisms during the degradation of polymers in the soil environment [7,39,40]. For this purpose, the OXI TOP technique was applied, and the determined BOD results (shown in Figure 6) confirmed the biodegradability of all the tested materials. However, their biodegradation rate depends on the characteristics of the component. The BOD values measured for all starch samples were repeatedly greater than those in the control (endogenous soil respiration). The modified starch films were characterized by a greater tendency to biodegrade up to the 12th day, after which the process slowed.

The NS/CA film was less susceptible to decomposition in the soil than the NS/QA sample. Mixtures containing both acids and DS were more resistant to biodegradation than samples without dialdehyde starch. The highest oxygen consumption was observed in the presence of NS/QA (with a BOD maximum of 1800 mg O_2_/kg of soil after 12 days of incubation), and the lowest value of this parameter was observed in the presence of NS/CA (with a BOD_max_ of ~1000 mg O_2_/kg of soil after 6 days of incubation).

The addition of DS decreased the BOD of starch-based films compared with the BOD of analogous films without this additive. This result confirmed the inhibition of biodegradation by DS cross-linking, which agrees with the literature [8,41,42].

## 4. Conclusions

The present research aimed to synthesize and characterize new ecological biomaterials dedicated to the packaging industry. Biodegradable starch-based films with dialdehyde starch and natural antioxidants (quinic and caffeic acid) were obtained by solution casting.

The mixtures varied in terms of their mechanical properties (i.e., they were better for blends with dialdehyde starch and quinic acid and the worst for films with caffeic acid (NS/DS/QA ≈ NS/QA ≫ NS/CA)) and antioxidant properties (NS/CA ≫ NS/QA). Caffeic and quinic acid promote the rigidity of starch films, while dialdehyde starch affects their stretchability. Spectroscopic characteristics, thermal behavior, and limited water vapor permeability through the membranes confirmed the mutual interactions and cross-linking between components, in which dialdehyde starch is particularly involved. The proposed starch modification increased amorphization, as confirmed by a decrease in orderliness and hydrophobization. All the mixtures were biodegradable and characterized by enhanced water vapor barrier properties. The moisture permeability and mechanical parameters are acceptable for packaging applications.

Starch-based films supplemented with antioxidants and enhanced with dialdehyde starch, i.e., vitamins (tocopherols, carotenes, carotenoids, ascorbic acid) or lipids, are alternatives for packaging purposes where oxidation protection is necessary.

## Figures and Tables

**Figure 1 polymers-16-00958-f001:**
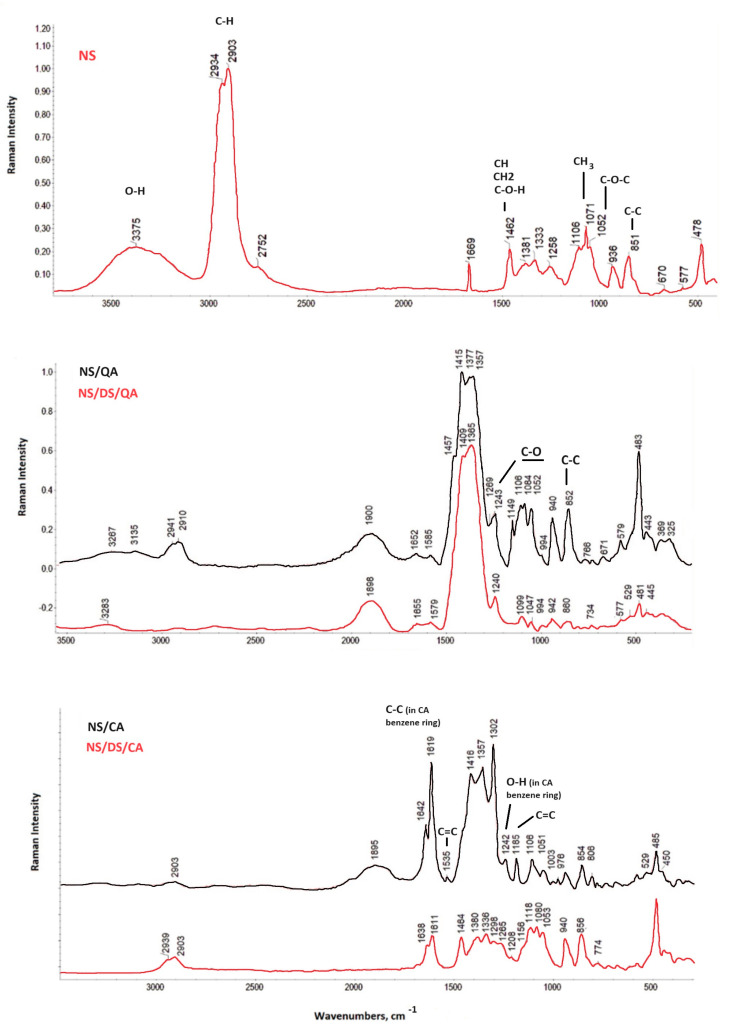
Raman spectra of a starch film (NS) and its blends with quinic acid with/without dialdehyde starch (NS/QA; NS/DS/QA) and with caffeic acid with/without dialdehyde starch (NS/CA; NS/DS/CA).

**Figure 2 polymers-16-00958-f002:**
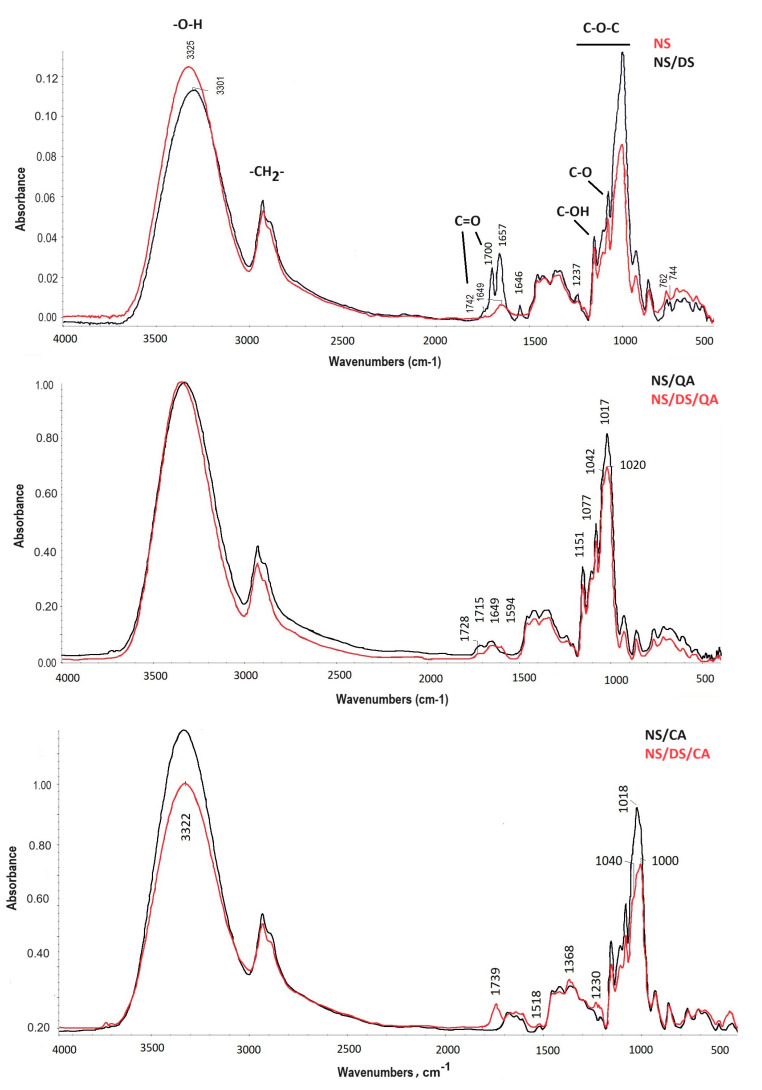
ATR-FTIR spectra of a starch film (NS) and its blends with quinic acid and/or dialdehyde starch (NS/QA; NS/DS/QA) and with caffeic acid and/or dialdehyde starch (NS/CA; NS/DS/CA).

**Figure 3 polymers-16-00958-f003:**
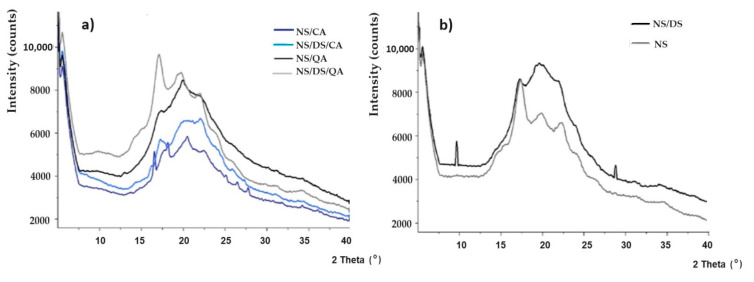
XRD spectra of (**a**) blends of native starch and caffeic/quinic acid (NS/CA, NS/QA) and native starch/dialdehyde starch and caffeic/quinic acid (NS/DS/CA, NS/DS/QA); (**b**) native starch (NS) and native starch/dialdehyde starch blend (NS/DS) [7].

**Figure 4 polymers-16-00958-f004:**
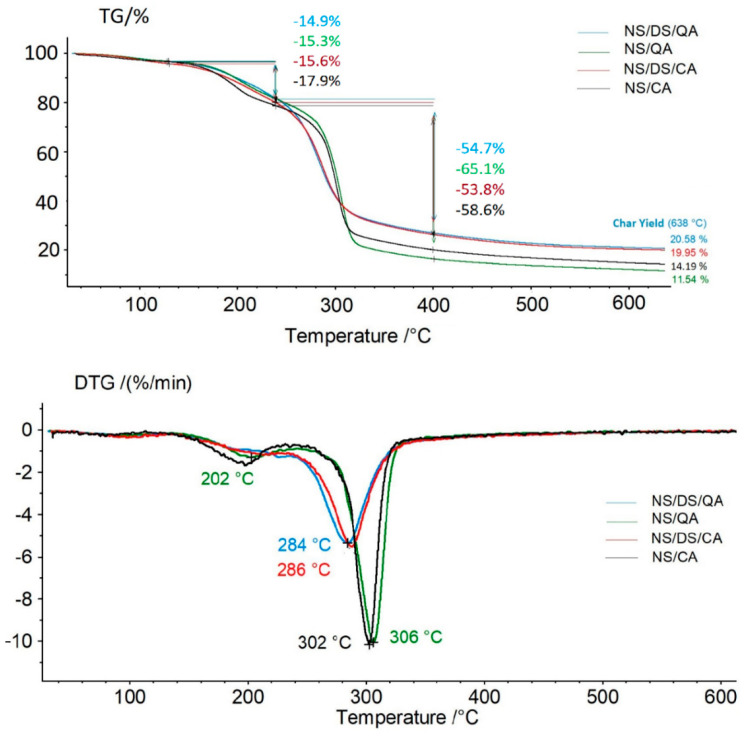
TG and DTG curves of “NS/QA”, “NS/DS/QA”, NS/CA”, “NS/DS/CA”.

**Figure 5 polymers-16-00958-f005:**
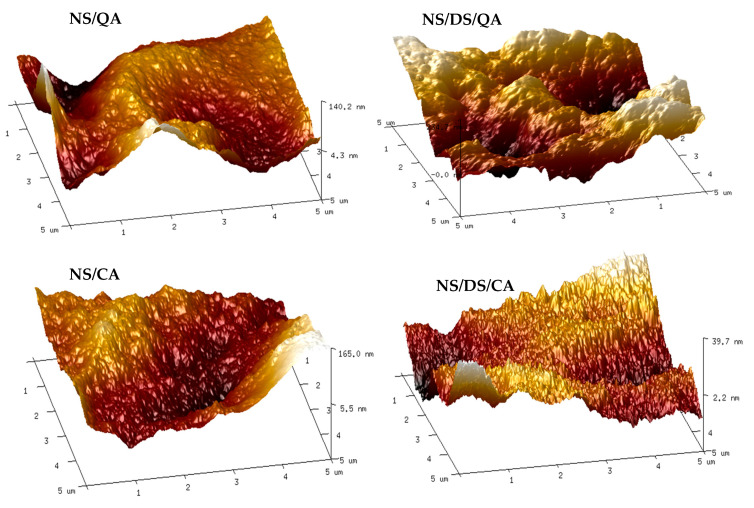
AFM images of the NS/QA, NS/DS/QA, NS/CA, and NS/DS/CA films.

**Figure 6 polymers-16-00958-f006:**
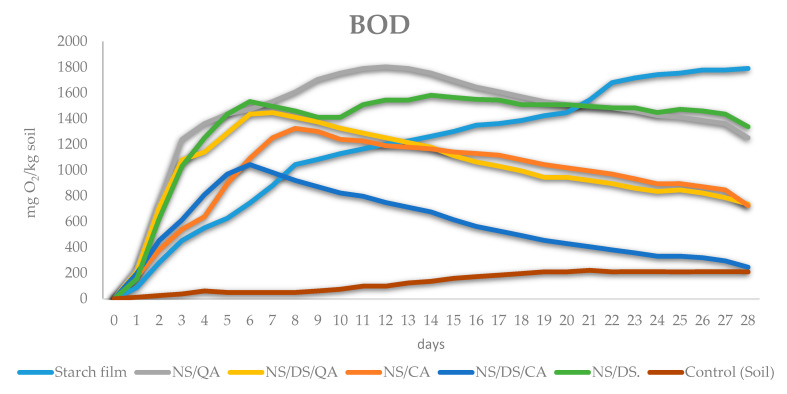
Biodegradation of starch films. Biological oxygen demand (BOD) during starch film biodegradation in soil.

**Table 1 polymers-16-00958-t001:** The parameters: the degree of crystallinity (X_c_, %), Young’s modulus (E, MPa), stress at break (υ_b_, MPa), elongation at break (ε_b_, %), water vapor transmission rate after 7 days (WVTR, g/m^2^·h), average roughness (R_a_, nm), root mean square roughness (R_q_, nm) and DPPH assay results (%) for the starch (NS), starch/dialdehyde starch (NS/DS), starch/caffeic acid (NS/CA), starch/dialdehyde starch/caffeic acid (NS/DS/CA), starch/quinic acid (NS/QA), and starch/dialdehyde starch/quinic acid (NS/DS/QA) films.

Sample	Crystallinity Degree (X_c_, %)	Young’s Modulus (E, MPa)	Stress at Break (υ_b_, MPa)	Elongation at Break (ε_b_, %)	Water Vapor Transmission Rate (WVTR, g/m^2^·h)	R_a_ (nm)	R_q_ (nm)	DPPH Assay (%)
NS	22.62	189 ± 4	6.7 ± 1.1	55.4 ± 2.1	7.78	3.90	5.5	0
NS/DS	9.35	144 ± 13	4.5 ± 0.56	91.3 ± 6.3	7.00	-	-	3.05
NS/CA	14.52	226 ± 34	5.4 ± 1.16	4.02 ± 5.8	7.17	27.4	33.0	80.5
NS/DS/CA	13.45	380 ± 46	2.6 ± 0.60	39.0 ± 4.5	6.65	7.7	9.6	69.1
NS/QA	13.73	383 ± 32	8.3 ± 1.04	49.2 ± 7.0	6.96	23	28.7	4.7
NS/DS/QA	15.51	366 ± 80	7.0 ± 0.47	52.3 ± 3.8	6.48	24	32.4	7.3

**Table 2 polymers-16-00958-t002:** Physical properties of selected synthetic polymers, biopolymers, and composites for the packaging industry [27,28,29,30,31,32,33].

Properties			Polymer/Composite								
	Poly(vinyl chloride) PVC [27]	Polypropylene (PP) [27]	High Density Polyethylene (HDPE)/ / Low-Density Polyethylene (LDPE) [27]	Polyhydroxy- Butyrate PHB [27]	Polylactide (PLA) PLA/Chitosan [27,30]	Polycapro- Laktone (PCL) [27,29]	Starch /Starch -Antioxidant Extract of Phenolic Acids [30,32]	PCL/Starch [27,33]	PCL/Starch/ Chitosan [27]	Chitosan/ Starch [27,28,29]	Chitosan/ Starch/ Thyme Extract [32]
Tensile strength (MPa)	4255	27–98	19–31/7–25	40	21–70/ 15–25	21–65	4.7–5.3/5.5–8.2 20/16.3–17.8	~25	18–28	34–40	11
Young’s Modulus (MPa)	2800	1500–2500	980/0.15–0.34	3500–4000	350–3800/ <1000	210–440	55–79/223–580 -	<80	78–100	-	82
Elongation at break (%)	10–20	>50%	840/-	5.0–8.0	2.5–6/ 125–200	300–1000	25–26/1.2–15 75/68.8–69.9	~500	20–180	28–61	36
WVTR (g/m^2^⋅day)	~1.1	3.9	4.6–8.0/16–23 (37 °C, 90%RH)	-	PLA: 333–1100 PLA/Chitosan: 16/42–54	~0.04	-	~0.08 (g/m^2^⋅h)	0.16–0.21 (g/m^2^⋅h)	Chitosan/starch: 46–53 (g/m^2^⋅h) (Chitosan: 4.8 )	-

## Data Availability

Data are contained within the article.

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
