# Peer review of "Eco-Friendly, Biodegradable Starch-Based Packaging Materials with Antioxidant Features"

_polymers, 2024, doi:10.3390/polym16070958_

Round 1
Reviewer 1 Report
Comments and Suggestions for Authors
The author presented a research article reporting characterization of starch films enhanced with dialdehyde starch and antioxidants. The article is generally well-written, but the clarity of data presentation and discussion should be improved. Overall, the author should explain the observation or readout from the characterization assays, and the reason causing the difference among the films tested, rather than only summarizing the data. For example, what is the reason for the different effect of dialdehyde starch on different acids regarding their mechanical properties? I would also suggest the author to address the following comments.
-
Why is the stress at break of NS/DS/CA film less than NS/CA in Table 1?
-
The Raman spectra and ATR-FTIR spectra in Figure 1 and 2 should be annotated with the bond corresponding to the peaks.
-
In some Figures, the Y axis is missing, eg. Figure 2, Figure 4.
-
Typo in Figure 5 caption: “AMF” should be “AFM”
-
What is the thickness of each film and how does that affect mechanical properties?
Minor editing of English language required
Author Response
Dear Reviewer,
Thank you very much for taking the time to review my manuscript. I have corrected the manuscript with these in mind. Please find the detailed responses below and the corresponding corrections in the resubmitted files I hope that none of the comments were left unanswered.
Comment 1: Why is the stress at break of NS/DS/CA film less than NS/CA in Table 1?
Response 1:
Thank you for this question. I also thought about it for a long time. The explanation that seems most likely to me, and which I also included in the manuscript, is as follows:
Lines 394-406:
“The effect of dialdehyde starch on NS/acid mixtures is markedly different, which may result from differences in homogeneity and the opportunities to new bond formation. The films of starch modified with CA were characterized by the most significant drop in mechanical parameters. The caffeic acid molecule may constitute a particular steric hindrance and limit the possibility of intermolecular hydrogen bond formation with starch, resulting the lowering the films` strength and stiffness. It may its consequence in phase-separation between blend component. The NS/DS/CA forms the most brittle film, characterized by the highest sensitivity to break (when compared with NS/CA and blends with quinic acid, which forms a more rigid material with higher fracture toughness). It indicates some difficulties in inter-molecular starch-caffeic acid-DS- binding, whereas intra-molecular hydrogen bonding in starch molecules may be preferable.”
Comments 2: The Raman spectra and ATR-FTIR spectra in Figure 1 and 2 should be annotated with the bond corresponding to the peaks.
Response 2: Obydwa rysunki zostały uzupełnione zgodnie z sugestią.
Comments 3,4: In some Figures, the Y axis is missing, eg. Figure 2, Figure 4. Typo in Figure 5 caption: “AMF” should be “AFM”
Response 3,4: Thank you for pointing this out. The figures have been completed. The caption of Figure 5 has also been corrected.
Comment 5: What is the thickness of each film and how does that affect mechanical properties?
Response 5: The average film thickness was 0.176 ± 0.023 mm) – this information is now provided in the line 152
Thank you again for the review. I hope I have improved the manuscript in accordance with all your valuable suggestion.
Best Regards,
Dagmara Bajer
Reviewer 2 Report
Comments and Suggestions for Authors
English needs to be improved further
Author Response
Dear Reviewer,
Thank you very much for taking the time to review my manuscript. I have corrected the manuscript with these in mind. Please find the detailed responses below and the corresponding corrections in the resubmitted files I hope that none of the comments were left unanswered.
Comment 1. The abstract is very concise and premature. It should be improved by adding some
information about the motivation and significance of the research, as well a sentence for future
directions.
Response 1: The abstract has been expanded in accordance with the proposal.
Comment 2. The introduction could be made more focused and concise by avoiding unnecessary
details and repetitions. For example, the paragraph on quinic acid and caffeic acid could be
shortened and merged with the previous paragraph on dialdehyde starch. Moreover, the
introduction should also clearly state the research gap, the research questions, and the hypotheses
to be tested in the study.
Response 2: Thank you very much for your valuable comments. I modified the Introduction to make it more precise and legible.
Comment 3. The article focuses on the properties and applications of starch-based packaging
materials, but does not provide any comparative analysis with other biodegradable materials, such
as cellulose, chitosan, or polylactic acid.
Response 3: I completely agree with this comment. This was an oversight that I corrected in the text: lines: 410-415, 437-450 and Table 2.
Comment 4. The results should be presented in a logical and coherent order, and the discussion
should be integrated with the results, rather than separated into different paragraphs
Response 4: I removed the separate paragraphs, leaving the results integrated in one Paragraph 3. ‘Results and discussion’
Comment 5: The figures and tables should be referred to in the text by their numbers, not by
expressions such as “below” or “above”. The figures and tables should also be clear and consistent,
and they should avoid unnecessary elements, such as grid lines, borders, and legends.
Comment 6. The axes labels, units, symbols, and fonts should be legible and uniform, and the
colors in plots should be distinguishable and meaningful. For instance Figure 1, 2 and 3 are not
legible.
Response 5 and 6: The Figures were corrected, Formulas were numbered 1-3 and referred to in the text by their numbers.
Comment 7. The conclusions should be concise and clear, and should summarize the main
findings and contributions of the study. The conclusions should also address the research questions
and hypotheses, and they should highlight the limitations and future directions of the research.
Response 7: I modified the conclusions only slightly to avoid their excessive expansion and repetition. They are precise, and I hope they are acceptable.
Thank you again for the review. I hope I have improved the manuscript in accordance with all your valuable suggestion.
Best Regards,
Dagmara Bajer
Reviewer 3 Report
Comments and Suggestions for Authors
The manuscript titled “Eco-friendly, biodegradable starch-based packaging materials with antioxidant features” submitted by Bajer Dagmara reports starch based films for potential application in packaging. The concept of this study is promising. However there are some queries which may be addressed before considering the manuscript for publication.
Please include some details about Optimization of concentrations, reaction conditions where necessary.
It would be of interest to readers if proposed Chemical linkages are described in the manuscript.
Section 2.10 the term “proper sample..” is not clear.
Author may consider describing reduction of Water vapor transmission rate in terms of statistical significance following enrichment/modification of NS.
For antioxidant properties please describe performance profile in comparison with a marketed product.
Comments on the Quality of English LanguageMinor editing is required
Author Response
Dear Reviewer,
Thank you very much for taking the time to review my manuscript. I have corrected the manuscript with these in mind. Please find the detailed responses below and the corresponding corrections in the resubmitted files I hope that none of the comments were left unanswered.
Comment 1: Please include some details about Optimization of concentrations, reaction conditions where necessary.
Response 1: Thank you for this suggestion. I tried to make the description of the presented experiments as accurate as possible; all conditions were described entirely, and by the procedure I used. I cannot be more specific, and I consider them complete.
Comments 2: It would be of interest to readers if proposed Chemical linkages are described in the manuscript.
Response 2: Yes, I Agree. These characteristics can be found in the text: lines 246-249; 254-257, 277-279, 314-323, and 346-348.
Comments 3: Section 2.10 the term “proper sample..” is not clear.
Response 3: The word “proper” was removed.
Comment 4: Author may consider describing reduction of Water vapor transmission rate in terms of statistical significance following enrichment/modification of NS.
Response 4: The Water vapour transmission rate (WVTR) is an essential parameter from the point of view of food packaging. In the case of the films I analyzed, the differences between WVTR values were only subtle, so it can be assumed that their permeability was comparable. Please find my comments here: lines 384-387 and the comparison with other materials: Table 2.
Comment 5: For antioxidant properties please describe performance profile in comparison with a marketed product.
Response 5: I agree with the reviewer's comment that the results of the obtained antioxidant properties were not compared with other data. However, since I did not find such data for commercial products, I compared it with other literature reports. It supplements the manuscript-lines 437-450.
Thank you again for the review. I hope I have improved the manuscript in accordance with all your valuable suggestion.
Best Regards,
Dagmara Bajer
Round 2
Reviewer 1 Report
Comments and Suggestions for Authors
Comments have been properly addressed.
Comments on the Quality of English LanguageMinor editing is required.
Author Response
Dear Reviewer,
Thank you very much for your valuable comments on my manuscript. I am attaching a revised version with language and editing corrections that align with your suggestions.
Best regards,
Dagmara Bajer